# Prodigiosin as an Algicidal Agent: Inhibition of Pigment Accumulation and Photosynthetic Efficiency of Cyanobacteria Involved in Algal Blooms

**DOI:** 10.3390/microorganisms13112569

**Published:** 2025-11-11

**Authors:** Chaobo Zhang, Chengshuai Xu, Zhenxia Zhu, Xiu Zhang, Zhaoan Shao, Zhenhui Yu, Zhangdi Zheng, Yijie Wang, Yadong Wang, Yujie Chen, Wei Xu, Jie Cheng

**Affiliations:** 1State Key Laboratory of Macromolecular Drugs and Large-Scale Preparation, School of Pharmaceutical Sciences and Food Engineering, Liaocheng University, Liaocheng 252000, China; zhangchaobo@lcu.edu.cn (C.Z.); 17661802272@163.com (Z.Z.); 15216501768@163.com (X.Z.); 18754003991@163.com (Z.S.); 17862565582@163.com (Z.Y.); zzd04141998@163.com (Z.Z.); 15069532882@163.com (Y.W.); 15206642957@163.com (Y.W.); sdh_cyj@163.com (Y.C.); xuwei@lcu.edu.cn (W.X.); 2Sino-Danish College, University of Chinese Academy of Sciences, Beijing 101408, China; chengshuaixu9926@126.com

**Keywords:** cyanobacterial blooms, algicidal effect, photosynthetic efficiency, prodigiosin, *M. aeruginosa*, *Anabaena* sp.

## Abstract

Eutrophication facilitates the proliferation of cyanobacteria, ultimately leading to the formation of harmful cyanobacterial blooms. Prodigiosin, known for its algicidal properties, presents significant potential for application in water pollution remediation. This study aims to identify and characterize a novel strain with superior prodigiosin production capabilities and to elucidate the algicidal mechanism of prodigiosin against *Microcystis aeruginosa* and *Anabaena* sp. by assessing the photosynthetic responses of algal cells in the presence of prodigiosin. The findings revealed the isolation and identification of a new strain, ZC52, classified as *Serratia marcescens*. The optimal medium composition was determined to be 20.0 mL·L^−1^ glycerol, 15.0 g·L^−1^ beef bone peptone, 15.0 g·L^−1^ magnesium sulfate heptahydrate, 0.15 g·L^−1^ corn dry powder, and 0.250% tyrosine, resulting in a 47.40% increase in prodigiosin yield, thereby achieving a production level of 7.644 g·L^−1^. Moreover, the algicidal activity exhibited a concentration-dependent relationship, with 10.0 mg·L^−1^ of prodigiosin leading to approximately 53.25% and 30.44% inhibition of chlorophyll a content within 24 h, demonstrating the potential of prodigiosin as an effective algicidal compound. Meanwhile, exposure to 10.0 mg·L^−1^ of prodigiosin resulted in reductions of 46.88% and 21.02% in the Fv/Fm values of *M. aeruginosa* and *Anabaena* sp., respectively. Our results indicated that prodigiosin can inhibit the accumulation of photosynthetic pigments and significantly diminish algal photosynthetic efficiency. This study not only identifies valuable microbial resources for prodigiosin production but also provides a theoretical framework and empirical evidence to support the scientific management of cyanobacterial blooms.

## 1. Introduction

Prodigiosin is a naturally occurring compound characterized by its three pyrrole rings and is classified within the prodiginine family. It serves as a biological pigment predominantly synthesized by microorganisms and demonstrates potential for various applications, including antimicrobial, algaecidal, antimalarial, antiprotozoal, and anticancer activities [1]. Due to its broad spectrum of biological activities and significant commercial value, there is increasing interest in optimizing prodigiosin production. The primary synthetic pathways for prodigiosin encompass chemical synthesis and microbial fermentation. Chemical synthesis involves multiple steps and often encounters technical challenges, such as limited structural specificity and complex processes, which impede large-scale production [2]. Conversely, microbial fermentation presents advantages such as cost-effectiveness, mild reaction conditions, and the absence of secondary pollution, positioning it as the preferred method for future industrialization [3].

In the context of prodigiosin production, a wide array of bacterial genera from both marine and terrestrial environments have been identified as potential candidates for synthesizing this compound. Notable examples include *Serratia marcescens*, *Serratia puccinata*, *Pseudomonas aeruginosa*, and various actinomycetes, with *S. marcescens* being the most extensively studied species [4,5,6,7]. In *Serratia* species, the biosynthesis of prodigiosin is regulated by the prodigiosin biosynthesis gene cluster, known as the pig cluster. This biosynthetic process involves enzymatic condensation, where monomeric pyrroles undergo chemical bonding to form multipyrrolic compounds. Despite advancements, the fermentation-based production of prodigiosin encounters several challenges, such as low pigment stability and suboptimal production efficiency. The significant variability in prodigiosin yield is largely due to differences in bacterial strains, medium composition (such as carbon and nitrogen sources), fermentation temperature, pH, and aeration [8,9,10,11,12]. Among these factors, the types of carbon and nitrogen sources, along with the specific bacterial strains employed, exert the most pronounced influence on both pigment composition and yield. Ensuring a consistent supply of prodigiosin for various biotechnological applications necessitates the development of bioprocesses capable of achieving high titers. Consequently, selecting novel strains and optimizing the medium composition and cultivation conditions are crucial to this objective.

The contamination of aquatic environments has become increasingly severe due to rapid industrial and agricultural development. Harmful algal blooms (HABs) are a significant consequence of marine pollution and eutrophication, posing serious adverse effects on both human health and marine ecosystems [13]. Previous research has demonstrated that prodigiosin exhibits algicidal activity, indicating considerable potential for its application in water pollution remediation [14,15]. The algicidal mechanism of prodigiosin primarily involves the disruption of algal cell membrane integrity, suppression of biological macromolecular substances synthesis, induction of reactive oxygen species production, and activation of the antioxidant systems [13,16]. These findings suggest new avenues for research into the algicidal biological activity of prodigiosin and offer novel insights into its application in environmental management. However, current studies on the algicidal function of prodigiosin remain in the preliminary stages, and its underlying photosynthetic mechanisms have not yet been fully elucidated.

This study concentrates on the isolation and identification of bacterial strains capable of producing prodigiosin, with efforts toward optimizing prodigiosin yield, providing both theoretical and experimental foundations for the large-scale production of prodigiosin. Simultaneously, using the predominant algal species involved in cyanobacterial blooms, namely *Microcystis aeruginosa* and *Anabaena* sp., as model organisms, a preliminary investigation was conducted into the mechanism by which prodigiosin inhibits algal growth through interference with photosynthesis. This research not only offers valuable microbial resources for prodigiosin production but also provides a theoretical framework and empirical data to support the scientific management of cyanobacterial blooms.

## 2. Materials and Methods

### 2.1. Isolation and Identification of Prodigiosin-Producing Strain

The soil samples used for the isolation of *S. marcescens* were collected from the perimeter soil surrounding East Lake at Liaocheng University, and LB agar medium comprising 10.0 g·L^−1^ tryptone, 5.0 g·L^−1^ yeast extract, 10.0 g·L^−1^ NaCl, and 15.0 g·L^−1^ agar was employed to isolate *S. marcescens*. The isolation procedure was conducted in a 50 mL sterile Erlenmeyer flask, with a soil sample to sterile water ratio of 1:10. The resulting mixture was subjected to serial dilution using a 10-fold gradient to achieve bacterial suspensions at concentrations of 10^−5^, 10^−6^, and 10^−7^. Subsequently, the bacteria were transferred from the Erlenmeyer flask to an LB agar plate using an inoculation loop. The plate was incubated at 30 °C for 24 h until red colonies appeared on the medium. A single colony was then selected and cultivated in LB liquid medium at a shaking speed of 200 rpm for 48 h.

The *S. marcescens* strain was identified through morphological and molecular biology methods. Morphological observation of the screened strains was conducted by macroscopic features, such as colony morphology, color, and texture, as well as microscopic features such as cell morphology under a microscope. Molecular biology identification was performed by 16S rDNA gene amplification. To amplify the 16S rDNA gene, genomic DNA extracted from the newly isolated prodigiosin-producing strain underwent polymerase chain reaction analysis utilizing universal primers (27F: AGAGTTTGATCMTGGCTCAG; 1429R: GGTTACCTTGTTACGACTT), as referenced in prior studies [17,18]. The PCR product obtained was subsequently purified and confirmed through sequencing by Sangon Biotech Co., Ltd. (Shanghai, China) The resulting sequence was aligned with existing sequences in GenBank using the BLAST algorithm (https://blast.ncbi.nlm.nih.gov/Blast.cgi, accessed on 5 November 2025), and a phylogenetic tree was constructed employing the neighbor-joining method with MEGA_X_10.2.6 [19].

### 2.2. Construction of Calibration Curve for Prodigiosin

The newly isolated prodigiosin-producing strain, designated as ZC52, was pre-cultured in 50.0 mL centrifuge tube with 10.0 mL basal medium comprising 20.0 mL/L glycerol (Shandong Keyuan Biochemical Co., Ltd., Heze, China), 15.0 g·L^−1^ bovine bone peptone (Jinan New Materials Industrial Park, Jinan, China), and 5.0 g·L^−1^ magnesium sulfate heptahydrate (Shandong Keyuan Biochemical Co., Ltd., Heze, China) at 30 °C with rotary shaking (200 rpm) for 48 h. The fermentation broth was centrifuged at 8000 rpm for 5 min, the supernatant was discarded, and the bacterial precipitate was washed three times with equal volumes of phosphate-buffered solution. The red pigment was extracted by adding equal volumes of ethyl acetate or acidic methanol comprising 4.0 mL of 1.0 mol·L^−1^ hydrochloric acid and 96.0 mL of methanol, and the extraction solutions were analyzed to determine their absorption spectra and to find out the maximum absorption using a spectrophotometer, with a scanning wavelength ranging from 300 nm to 1000 nm. Subsequently, crude extraction of prodigiosin from the fermentation broth was carried out based on previous literature [20], and a standard curve for the crude extract of prodigiosin was constructed.

### 2.3. Batch Prodigiosin Production Procedure

ZC52 was pre-cultured in a 50.0 mL centrifuge tube with 10.0 mL basal medium, and the cells obtained were harvested by centrifugation (8000 rpm, 10 min), washed, and resuspended with fresh basal medium. The batch tests were performed in a 250.0 mL sterile Erlenmeyer flask filled with 50.0 mL basal medium, and the strain was cultivated with 10% of the inoculation amount at 30 °C with a speed of 200 rpm for 72 h. The prodigiosin production of ZC52 at different time intervals (12, 24, 36, 48, 60, and 72 h) was obtained by the calibration curve of prodigiosin.

To optimize the production performance of prodigiosin by ZC52, three factors—carbon sources, nitrogen sources, and inorganic salts—were systematically investigated. Specifically, glycerol was utilized to examine the impact of varying carbon source concentrations on prodigiosin production, with glycerol concentrations ranging from 5.0 to 25.0 mL·L^−1^. Meanwhile, the influence of nitrogen sources was assessed by introducing bovine bone peptone at concentrations of 20.0 g·L^−1^, 25.0 g·L^−1^, 30.0 g·L^−1^, 35.0 g·L^−1^, 40.0 g·L^−1^, and 45.0 g·L^−1^. Additionally, the response of prodigiosin production in ZC52 to different initial concentrations of magnesium sulfate heptahydrate was evaluated at 1.0 g·L^−1^, 5.0 g·L^−1^, 10.0 g·L^−1^, 15.0 g·L^−1^, and 20.0 g·L^−1^. Meanwhile, the effects of varying concentrations of corn dry powder and tyrosine on prodigiosin production were also examined, with corn dry powder concentrations set at 0.0 g·L^−1^, 0.15 g·L^−1^, 0.3 g·L^−1^, 0.45 g·L^−1^, and 0.6 g·L^−1^, and tyrosine concentrations at 0.0%, 0.125%, 0.25%, 0.375%, and 0.5%, respectively. All experiments were performed in triplicate. Based on the above single-factor experiments, L_16_(4^5^) was designed through an orthogonal experiment to investigate the interaction between various factors (Table 1) and analyze the key influencing factors that affect the production of prodigiosin by ZC52.

### 2.4. Algal Culture Preparation and Algicidal Activity of Prodigiosin

*M. aeruginosa* (FACHB-915) and *Anabaena* sp. (FACHB-82) were purchased from the Freshwater Algae Culture Collection of the Institute of Hydrobiology of the Chinese Academy of Sciences and preserved by the Modern Biotechnology and Environmental Protection Research Group at Liaocheng University. The algal strains were cultured in BG-11 medium and incubated within a controlled light culture chamber. The cultivation parameters were meticulously maintained as follows: a temperature of 25 ± 1 °C, a light intensity of 3000 lux, a light/dark cycle of 14 h of light followed by 10 h of darkness, and a rotational speed of 120 rpm.

Prodigiosin, dissolved in acidic methanol, was introduced to the algal cultures during the exponential growth phase, achieving a final concentration of either 5.0 mg·L^−1^ or 10.0 mg·L^−1^, and acidic methanol, serving as a control, was also added into algal cultures with an equal volume as prodigiosin. The cell cultures were propagated in autoclaved 250 mL Erlenmeyer flasks, each containing 100 mL of BG-11 medium, at a consistent temperature of 25 °C. Microalgal growth was assessed by quantifying the chlorophyll a content in liquid cultures, as chlorophyll a is a crucial indicator when direct cell counts are not feasible [21]. To determine the content of photosynthetic pigments, 4.0 mL of algal cells were collected via centrifugation and subsequently washed with phosphate-buffered solution (PBS, 0.1 M, pH 7.4). An equal volume of 95% ethanol was added to the cell pellets, and the photosynthetic pigments were extracted at 4 °C. The extract solutions were centrifuged at 12,000 rpm for 5 min, and the absorbance of the supernatant was measured at wavelengths of 665 nm, 649 nm, and 470 nm. The concentrations of chlorophyll *a* and carotenoids were calculated based on established literature methods [22].

### 2.5. Analysis of Photosynthetic Inhibition Mechanisms of Prodigiosin on Algae Growth by Chlorophyll a Fluorescence Assay

Fluorescence parameters serve as reliable indicators of a sample’s physiological state, encompassing aspects such as growth, stress exposure, and general health [23]. The inhibitory effects of prodigiosin on photosynthetic activity in algal blooms were assessed by evaluating chlorophyll a fluorescence transient, measured with a chlorophyll fluorometer (AquaPen-C AP110-C, Photon Systems Instruments, Drásov, Czech Republic), and the JIP transients were quantified following exposure to prodigiosin. Prior to measurement, microalgae cultures (10.0 mL) treated with prodigiosin were centrifuged at 8000 rpm for 5 min. The algal cells were then resuspended in fresh BG-11 medium to standardize the chlorophyll a concentration across treatment groups. Samples were preconditioned in darkness for a minimum of 20 min before fluorescence parameter measurements. The formulae and glossary of terms pertinent to the JIP-test and the analysis of the O–J–I–P fluorescence transient are documented in [24]. Each experimental procedure was conducted in triplicate to ensure reliability.

### 2.6. Data Processing

The data were analyzed using univariate analysis of variance (ANOVA) and Duncan’s multiple comparison test. Statistical significance was established at a 5% level (*p* < 0.05). SPSS 19.0 statistical software (SPSS Inc., Chicago, IL, USA) was utilized for data analysis, and diagrams were plotted with GraphPad Prism 9.0 (San Diego, CA, USA).

## 3. Results

### 3.1. Identification of Newly Isolated S. marcescens Strain

The soil sample was inoculated onto LB agar plates, followed by incubation at 30 °C for a duration of 1 to 2 days. Post-cultivation, distinct red monoclonal colonies became visible on the surface of the LB solid medium. These colonies were characterized by a smooth, moist surface and well-defined margins (Figure 1A). Microscopic examination revealed that the colonies comprised short rod-shaped or spherical cells (Figure 1B). The 16S rDNA sequence was amplified using the genomic DNA of the strain as a template, employing universal primers 27F/1492R. Agarose gel electrophoresis analysis displayed a band of approximately 1500 bp (Figure 1C), aligning with the expected size. The target band was subsequently purified and dispatched to Sangon Biotech Co., Ltd. for sequencing. The resulting 16S rDNA sequence was compared against the NCBI database using BLAST. Twelve model strains exhibiting high homology were selected and downloaded for the construction of a phylogenetic tree using MEGA X (Figure 1D). The findings indicated that the strain shares the closest phylogenetic relationship with *S. marcescens*, leading to its designation as *S. marcescens* ZC52.

### 3.2. Prodigiosin Characterization and Preliminary Yield Evaluation in S. marcescens ZC52

Prodigiosin exhibits limited solubility in aqueous solutions but demonstrates significant solubility in organic solvents, including methanol and ethyl acetate (Figure 2A). In the present study, the red pigment was extracted from the fermentation broth of *S. marcescens* ZC52 utilizing acidified methanol and ethyl acetate, respectively. A comprehensive wavelength scan of the pigment extract was conducted (Figure 2B), revealing a maximum absorption wavelength at 535 nm. These findings align with previously documented characteristics of prodigiosin, thereby preliminarily confirming the potential of *S. marcescens* ZC52 to synthesize this compound.

To facilitate the rapid quantification of prodigiosin content in *S. marcescens* ZC52, a standard curve was established, plotting the concentrations of crude prodigiosin extract (g·L^−1^) against absorbance values (OD_535_) (Figure 2C). The results indicated a robust linear correlation between prodigiosin concentrations and OD_535_ values within a defined range. The linear regression equation was determined to be Y = 0.0296X, with a coefficient of determination (R^2^) of 0.9999. Furthermore, the kinetics of prodigiosin production by *S. marcescens* ZC52 in a basal medium are depicted in Figure 2D. The production of prodigiosin with a yield of 5.106 g·L^−1^ was observed at 72 h, underscoring its significant potential for industrial applications.

### 3.3. Optimizations of Medium Composition for Improving the Prodigiosin Yield

The classical optimization method, which involves altering one independent variable while maintaining other constants to assess their individual effects on production, was employed in this study. This approach was used to determine the optimal levels of various parameters to maximize prodigiosin production. The study investigated variables such as carbon sources, nitrogen sources, and inorganic salts (Figure 3). The findings revealed that the addition of glycerol at a concentration of 15 mL·L^−1^ led to an intracellular prodigiosin accumulation of 5.782 g·L^−1^, marking a 13.24% increase relative to the control group. Additionally, bovine bone peptone was found to significantly influence prodigiosin synthesis. As the concentration of bovine bone peptone increased, the intracellular prodigiosin accumulation exhibited an initial increase followed by a decrease. Among the conditions tested, the incorporation of 25.0 g·L^−1^ bovine bone peptone was most favorable for prodigiosin production by *S. marcescens* ZC52, resulting in a yield of 6.070 g·L^−1^, which represents an 18.88% enhancement over the control. Furthermore, the study examined the effects of varying concentrations of magnesium sulfate heptahydrate, with results indicating that a concentration of 10.0 g·L^−1^ produced the highest level of prodigiosin, amounting to 1.27 times that of the control group. These findings substantiate that the concentrations of medium components significantly influence prodigiosin synthesis in *S. marcescens* ZC52.

The incorporation of exogenous additives emerges as a promising strategy to enhance the biosynthesis efficiency of prodigiosin. As illustrated in Figure 3, the addition of 0.3 g·L^−1^ of corn dry powder resulted in a 32.86% increase in prodigiosin production by *S. marcescens* ZC52 compared to the control group. Furthermore, tyrosine was identified as a significant factor influencing prodigiosin synthesis, with a supplementation level of 0.25% leading to a peak yield of 7.526 g·L^−1^. Consequently, the addition of both corn dry powder and tyrosine markedly affects the accumulation of prodigiosin in *S. marcescens* ZC52.

Based on single-factor experiments, the concentrations of these five factors were further refined using an orthogonal experimental design, with the results detailed in Table 2. The optimal combination for maximizing prodigiosin production was identified as A4B1C4D2E3, corresponding to glycerol 20.0 mL·L^−1^, beef bone peptone 15.0 g·L^−1^, magnesium sulfate heptahydrate 15.0 g·L^−1^, corn dry powder 0.15 g·L^−1^, and tyrosine 0.250%. Range analysis revealed that, among the five factors, bovine bone peptone exerted the most substantial influence on prodigiosin production, followed by tyrosine, glycerol, and corn dry powder, while magnesium sulfate heptahydrate had the least influence on prodigiosin accumulation. Under the optimal conditions, the prodigiosin yield reached 7.644 g·L^−1^, representing a 47.40% increase compared to that before optimization.

### 3.4. Algicidal Activity of Prodigiosin on M. aeruginosa and Anabaena sp.

Photosynthetic pigments are essential parameters for assessing microalgal photosynthesis, as they not only indicate the growth status of algal strains but also serve as bio-stress markers to evaluate damage under stressful conditions. Figure 4 presents the intracellular pigment accumulation in *M. aeruginosa* and *Anabaena* sp. when exposed to varying concentrations of prodigiosin. As depicted in Figure 4A, treatment with prodigiosin resulted in a significant reduction in the content of photosynthetic pigments (*p* < 0.05), with the inhibitory effect becoming more pronounced at higher prodigiosin concentrations. Specifically, at a prodigiosin concentration of 10.0 mg·L^−1^, the intracellular levels of chlorophyll *a* and carotenoids were measured at 0.539 μg·mL^−1^ and 0.186 μg·mL^−1^, respectively, corresponding to reductions of 53.25% and 50.13% compared to the control group.

Additionally, this study examined the impact of prodigiosin treatment on the accumulation of photosynthetic pigments in *Anabaena* sp. (Figure 4B). While exposure to 5.0 mg·L^−1^ prodigiosin did not result in significant changes in intracellular pigment content, treatment with 10.0 mg·L^−1^ prodigiosin markedly inhibited pigment accumulation. Compared to the control group, the 10.0 mg·L^−1^ prodigiosin treatment led to reductions of 30.44% and 29.00% in chlorophyll a and carotenoids content, respectively. Furthermore, under the same prodigiosin concentration, the inhibition rate of photosynthetic pigments in *Anabaena* sp. was markedly lower than that in *M. aeruginosa*, indicating that *M. aeruginosa* was more sensitive to environmental exposure of prodigiosin.

### 3.5. Photosynthetic Inhibition Mechanisms of Prodigiosin on Algae Blooms

#### 3.5.1. Effects of Prodigiosin on Chlorophyll Fluorescence Characteristics

Figure 5 presents the temporal analysis of OJIP curves under varying concentrations of prodigiosin. As depicted in Figure 5A, the OJIP curves for *M. aeruginosa* exhibited no significant alterations following a 24 h exposure to low concentrations of prodigiosin. Conversely, as prodigiosin concentrations increased, there was a notable upward trend in the J-phase and I-phase levels of the OJIP curve. This trend suggests a reduction in the efficiency of electron transfer from Q_A_ to Q_B_ within the photosynthetic electron transport chain, resulting in the excessive accumulation of Q_A_^−^ within the cells and subsequent cellular damage. In contrast, the OJIP curves for *Anabaena* sp. exposed to varying concentrations of prodigiosin retained a typical distribution without significant differences (Figure 5B), corroborating the heightened sensitivity of *M. aeruginosa* to environmental prodigiosin exposure.

#### 3.5.2. Effects of Prodigiosin on the Light Energy Conversion Efficiency and Energy Distribution in the PSII Reaction Center of *M. aeruginosa* and *Anabaena* sp.

Figure 6A illustrates the impact of prodigiosin exposure on the Fv/Fm values of the algal strains. Compared to the control group, exposure to 10.0 mg·L^−1^ prodigiosin resulted in reductions of 46.88% and 21.02% in the Fv/Fm values of *M. aeruginosa* and *Anabaena* sp., respectively. These findings suggest that prodigiosin stress induces irreversible inactivation of the PSII reaction centers in the algal cells. Meanwhile, the impact of varying concentrations of prodigiosin on energy absorption, capture, and transfer within the PSII reaction centers of algal strains was also investigated. As illustrated in Figure 6B, exposure to prodigiosin resulted in a marked increase in the ABS/RC and DIo/RC values in both *M. aeruginosa* and *Anabaena* sp. Specifically, relative to the control group, the ABS/RC values in the 10.0 mg·L^−1^ prodigiosin treatment group rose by 83.65% and 66.50%, respectively. And the DIo/RC values increased by 1.92-fold and 1.72-fold, respectively. These findings demonstrated that external stress diminished the photosynthetic efficiency of algal cells, prompting them to capture more energy to maintain vital functions and thereby stimulating intracellular energy accumulation. Meanwhile, a portion of the absorbed energy is dissipated as heat to prevent cellular damage from energy overloading, reflecting a self-protective mechanism employed by algal cells under stress conditions.

#### 3.5.3. Effect of Prodigiosin on the Photosynthetic Electron Transfer Efficiency of *M. aeruginosa* and *Anabaena* sp.

Mo reflects the net rate of closure of PSII reaction centers and is frequently employed to assess the reduction rate of Q_A_. Sm denoted the energy necessary for the complete reduction of Q_A_, thereby indicating the size of the plastoquinone (PQ) pool on the acceptor side of PSII reaction centers. As illustrated in Figure 7, exposure to prodigiosin resulted in significantly elevated Mo values in both *M. aeruginosa* and *Anabaena* sp. compared to the control group. Meanwhile, within the tested concentration range of prodigiosin, the Sm value exhibited an increase with rising prodigiosin concentrations. Following treatment with 10.0 mg·L^−1^ prodigiosin, the Sm values of the algal cells were 4.29-fold and 2.72-fold higher than those of the control group, respectively.

The effects of prodigiosin on the V_J_ and V_I_ values of *M.aeruginosa* and *Anabaena* sp. are illustrated in Figure 7. V_J_ reflects the electron transfer efficiency from Q_A_ to Q_B_, with a higher V_J_ value indicating lower electron transfer efficiency. V_I_ reflects the reduction of PQ and the reoxidation process of PSI. In this study, treatment with 10.0 mg·L^−1^ prodigiosin resulted in V_J_ and V_I_ values in *M. aeruginosa* that were 1.35-fold and 1.29-fold higher than those of the control group, respectively. However, no significant differences were observed in the V_J_ and V_I_ values of *Anabaena* sp. under prodigiosin treatment. Additionally, this study examined the impact of prodigiosin treatment on the ETo/ABS and ETo/TRo metrics of various algal strains. The ETo/ABS value represented the quantum yield of electron transport, whereas the ETo/TRo value denoted the probability that electrons captured by the PSII reaction center enter the electron transport chain. As illustrated in Figure 7, following 24 h of exposure to 10.0 mg·L^−1^ prodigiosin, the ETo/ABS and ETo/TRo values for *M. aeruginosa* decreased by 53.82% and 13.43%, respectively, compared to the control group. Meanwhile, treatment with 10.0 mg·L^−1^ prodigiosin resulted in a 17.18% reduction in the ETo/ABS value for *Anabaena* sp. relative to the control group.

## 4. Discussion

### 4.1. Enhancing Prodigiosin Production by Optimizing Medium Composition

Prodigiosin, a prominent compound within the 4-methoxypyrrolyldipyrrin family of natural products, exhibits a wide range of biological activities [25,26,27]. The microbial production of prodigiosin is hindered by challenges such as relatively low yields and high production costs, while its chemical synthesis involves intricate processes. Consequently, screening and excavating new strains with excellent prodigiosin production performance is extremely important for providing a valuable microbial resource for future prodigiosin production research. Previous studies have demonstrated that various microorganisms are capable of synthesizing this pigment, with *S. marcescens* exhibiting relatively high levels of prodigiosin production and being the most extensively investigated species. *S. marcescens* is commonly found in soil, water, and within plant and animal hosts. In recent years, strains capable of consistently producing the red pigment have also been isolated from unique environments, including air, blood, and compost [28,29]. Among these, *S. marcescens* strains isolated from soil environments are considered the most promising candidates for industrial-scale prodigiosin production [30,31,32]. In this study, a red pigment-producing microbial strain was isolated from the perimeter soil surrounding East Lake at Liaocheng University. Through microscopic observation, 16S rRNA gene sequencing, and phylogenetic analysis, the strain was identified as closely related to *S. marcescens* and was designated as *S. marcescens* ZC52.

Optimizing the production of prodigiosin by *S. marcescens* is essential for reducing production costs and expanding its potential applications. The initial approach to enhancing bacterial growth and product yield involves optimizing the culture medium through nutrient supplementation, thereby increasing both biomass and the concentration of the target metabolite. As a fundamental component of microbial metabolism, the carbon source is crucial for providing the energy necessary for cellular growth [33]. Previous studies have demonstrated that *S. marcescens* can utilize a diverse array of carbon sources, with glycerol identified as a particularly effective option for maximizing prodigiosin production [34,35]. In this study, *S. marcescens* ZC52 was cultured in varying concentrations of glycerol to assess its impact on prodigiosin production. The findings revealed that the highest prodigiosin yield, 5.782 g·L^−1^, was achieved in cultures supplemented with 15 mL·L^−1^ glycerol, corroborating results from previous research [35].

Moreover, the synthesis of pyrrole rings in prodigiosin necessitates a nitrogen source, with organic nitrogen sources proving significantly more advantageous for prodigiosin synthesis than inorganic ones [36,37]. In this study, the highest yield of prodigiosin (6.070 g·L^−1^) was achieved in a culture supplemented with 25.0 g·L^−1^ of bovine bone peptone as the exclusive nitrogen source. This finding aligns with several studies indicating that prodigiosin production is enhanced in cultures containing peptone [8,38]. Inorganic salts, serving as enzymatic cofactors, facilitate the biosynthesis of target compounds. Specifically, the Mg^2+^ ion plays a crucial role in maintaining intracellular acid-base equilibrium and promoting efficient nutrient utilization in bacteria [39]. Consequently, various concentrations of magnesium sulfate were tested to optimize prodigiosin production. The optimal concentration of magnesium sulfate was determined to be 10.0 g·L^−1^, resulting in the highest prodigiosin yield of 6.485 g·L^−1^.

Despite the fact that optimizing the basal medium can increase prodigiosin yield, researchers have also discovered that incorporating regulatory agents into the medium can further enhance production. For decades, culture media utilizing amino acids as carbon sources have been a standard approach to support and promote bacterial growth [40,41]. Previous research indicates that the production of prodigiosin relies on biosynthetic pathways that incorporate specific amino acids as key building blocks. Among them, histidine and proline, which possess pyrrole-like structures, serve as integral precursors for the synthesis of this pigment [42]. Other amino acids, including proline, glutamate, alanine, and serine, have also been demonstrated to stimulate production [43]. The incorporation of methionine, in particular, has been found to reduce the lag time of prodigiosin synthesis and enhance pigment yield [44]. Notably, previous research has confirmed that tyrosine supplementation is more effective than alanine or proline in promoting prodigiosin production [45]. In this study, tyrosine was added to the fermentation medium as a substrate to increase prodigiosin production, and the prodigiosin production was significantly affected by the addition of tyrosine. The results of the present study recorded maximum production of prodigiosin at 0.25%.

The commercial application of prodigiosin has been constrained by its high production costs, primarily attributed to the expensive growth substrates required. Consequently, identifying low-cost substrates represents a viable strategy to mitigate these production expenses. Various agricultural industrial wastes and by-products, such as corn dry powder, retain substantial nutrient content and can be employed to enhance prodigiosin synthesis by *S. marcescens* through the optimization of carbon source utilization efficiency. As anticipated, the use of corn dry powder, an economical substrate, resulted in a significant increase in prodigiosin production, with the highest yield observed at a concentration of 0.3 g·L^−1^. Building on single-factor experimental findings, an orthogonal experimental design was employed to explore the interactions among five key factors influencing the intracellular accumulation of prodigiosin in the ZC52 strain. The results demonstrated that under optimal fermentation conditions, prodigiosin yield reached 7.64 g·L^−1^, suggesting that the ZC52 strain possesses considerable potential for industrial applications.

### 4.2. Evaluation of the Algicidal Mechanisms of Prodigiosin

Eutrophication facilitates the proliferation of cyanobacteria, ultimately leading to harmful algal blooms [46,47]. Algicidal bacteria contribute significantly to the suppression of such blooms and are regarded as promising agents for bloom mitigation. Several algicidal compounds derived from these bacteria, which target *Microcystis* blooms, have attracted considerable scientific interest [48]. Recently, prodigiosin has also been proven effective in controlling harmful algal blooms, demonstrating broad-spectrum algicidal activity. Notably, it has been reported not only to regulate *Microcystis* blooms and inhibit microcystin production [48], but also to act as an algicide against *Heterosigma akashiwo* [13], which was a motile, flagellated eukaryotic alga from the stramenopiles lineage and was a common representative of harmful marine blooms. However, current studies on the algicidal function of prodigiosin remain in the preliminary stages, and its underlying photosynthetic mechanisms have not yet been fully elucidated.

In this study, the algicidal activity of prodigiosin against *M. aeruginosa* and *Anabaena* sp. was investigated. The results revealed a concentration-dependent inhibitory effect, where exposure to 10.0 mg·L^−1^ prodigiosin for 24 h led to reductions in chlorophyll *a* content of approximately 53.25% and 30.44%, respectively, indicating its potential as an algicidal agent. Under environmental stress, microalgae often trigger photosynthetic responses to mitigate adverse effects and initiate cellular repair. The chlorophyll *a* fluorescence transient, particularly analyzed via the JIP test, serves as a sensitive indicator of photosynthetic performance, reflecting disturbances in the PSII electron transport chain. Therefore, the OJIP transient was employed to elucidate the photosynthetic inhibition mechanism of prodigiosin on the tested cyanobacteria.

The chlorophyll *a* fluorescence transient (OJIP curve) is a valuable technique for assessing photosynthesis performance and plant stress status [49]. This curve represents the changes in chlorophyll fluorescence intensity over time, capturing the dynamics of photosystem II (PSII) during the initial seconds of illumination. The OJIP curve is characterized by distinct phases—O, J, I, and P—that reflect various stages of energy absorption, electron transport, and photochemical quenching within the photosynthetic apparatus [50]. Understanding these phases is crucial for interpreting the physiological status of plants under different environmental conditions. The OJIP curve is composed of distinct phases: the O–J phase, the J–I phase, and the I–P phase [51]. The O-point signifies the minimum fluorescence level observed when dark-adapted algal cells are initially exposed to light, serving as an indicator of the efficiency of light energy utilization. The J-point, observed at approximately 2 milliseconds, represents the electron transfer from the primary electron acceptor (Q_A_) to the secondary electron acceptor (Q_B_) within the PSII reaction center. The I-point, occurring at approximately 30 milliseconds, corresponds to the reduction of plastoquinone and the subsequent re-oxidation of Photosystem I (PSI). Finally, the P-point indicates the maximum fluorescence level when the PSII reaction centers are fully closed [52]. In this study, the continuous increase in J-phase and I-phase levels of the OJIP curve after treatment with 10 mg·L^−1^ of prodigiosin confirmed that there was significant irreversible efficiency of electron transfer to algal cells, resulting in the excessive accumulation of Q_A_^−^ within the cells and subsequent cellular damage.

The maximum photochemical efficiency (Fv/Fm) of PSII reaction centers was a crucial parameter for assessing the photochemical activity of PSII and predicting the stress status in plants [53]. Under non-stressed conditions, this value typically remains stable, showing minimal variation across different plant species or environmental conditions. However, when algal cells experience stress, the Fv/Fm value significantly decreases, providing a sensitive indicator of the extent of functional impairment in the photosynthetic system [54]. Our results indicated that prodigiosin had a significant consequence on the photosynthetic capacity of *M. aeruginosa* and *Anabaena* sp., and reductions of 46.88% and 21.02% in the Fv/Fm values were obtained when exposed to 10.0 mg·L^−1^ prodigiosin, indicating that the light energy conversion efficiency of *M. aeruginosa* and *Anabaena* sp. was inhibited. Previous studies have confirmed that the Fv/Fm value in *Phaeocystis globosa* decreased after prodigiosin treatment [16], which was consistent with our research findings.

Under typical physiological conditions, photosynthetic microorganisms harness light energy through active photosystem II (PSII) reaction centers to drive photochemical reactions. This process encompasses electron transport and coupled photophosphorylation, culminating in the production of assimilatory power, which facilitates carbon assimilation. However, under external environmental stress, PSII reaction centers undergo reversible inactivation, functioning as energy sinks that dissipate the majority of absorbed light energy as heat. Our finding that prodigiosin caused a significant increase in the ABS/RC of *M. aeruginosa* and *Anabaena* sp. revealed that the addition of prodigiosin inactivated the reaction centers, leading to an increase in light absorption capacity of the PSII unit reaction centers. It is postulated that the induction of Q_A_-non-reducing reaction centers allowed treated cells to absorb light but impeded Q_A_ reduction. Consequently, excess energy was likely liberated thermally, a notion corroborated by the elevated DIo/RC value [55]. Thus, an algal self-protection mechanism was likely triggered to mitigate photooxidative damage [56].

Finally, we investigated the effects of prodigiosin on the photosynthetic electron transfer efficiency of *M. aeruginosa* and *Anabaena* sp. Exposure to prodigiosin resulted in significantly elevated Mo and Sm values in both *M. aeruginosa* and *Anabaena* sp. compared to the control group, indicating that prodigiosin stress enhances the net rate of PSII reaction center closure and impedes electron transfer from Q_A_ to downstream electron acceptors (Q_B_, PQ, Cytb6f), thereby leading to an excessive accumulation of Q_A_^-^ within the cells. Meanwhile, treatment with 10.0 mg·L^−1^ prodigiosin resulted in V_J_ and V_I_ values in *M. aeruginosa* that were 1.35-fold and 1.29-fold higher than those of the control group, further demonstrating that prodigiosin treatment inhibits the electron transfer efficiency from Q_A_ to Q_B_ in algal cells and simultaneously suppresses the transfer of photosynthetic electrons from PQ to the electron acceptors of the PSI reaction center [57]. In addition, following 24 h of exposure to 10.0 mg·L^−1^ prodigiosin, the ETo/ABS and ETo/TRo values for *M. aeruginosa* decreased by 53.82% and 13.43%, respectively, compared to the control group. Meanwhile, treatment with 10.0 mg·L^−1^ prodigiosin resulted in a 17.18% reduction in the ETo/ABS value for *Anabaena* sp. relative to the control group. These results suggested that prodigiosin treatment diminished the proportion of excitons captured by the PSII reaction center that are utilized to drive electron acceptors beyond Q_A_ in the electron transport chain, as opposed to those used for Q_A_ reduction. Thus, prodigiosin treatment appeared to inhibit the photosynthetic electron transport efficiency of algal cells. Altogether, the decline in photosynthetic efficiency correlated with higher prodigiosin levels, indicating that inhibition likely occurred via disruption of electron transport in the studied cyanobacteria.

## 5. Conclusions

In this study, the ZC52 strain, characterized by its high prodigiosin production capability, was isolated and identified as *S. marcescens*. Various nutritional factors were systematically examined for their effects on prodigiosin production using an orthogonal experimental design. The optimal medium composition was determined to be 20.0 mL·L^−1^ glycerol, 15.0 g·L^−1^ beef bone peptone, 15.0 g·L^−1^ magnesium sulfate heptahydrate, 0.15 g·L^−1^ corn dry powder, and 0.250% tyrosine, which resulted in a 47.40% increase in prodigiosin yield, achieving a production level of 7.644 g·L^−1^. These findings suggest that the ZC52 strain, with its high prodigiosin production efficiency, represents a valuable microbial resource for future prodigiosin production research. Furthermore, the study assessed the photosynthetic inhibition mechanisms of prodigiosin on algal blooms using chlorophyll fluorescence analysis. Increased prodigiosin concentrations were found to inhibit photosynthetic pigment accumulation and significantly reduce algal photosynthetic efficiency, highlighting its substantial potential for application in water pollution remediation.

## Figures and Tables

**Figure 1 microorganisms-13-02569-f001:**
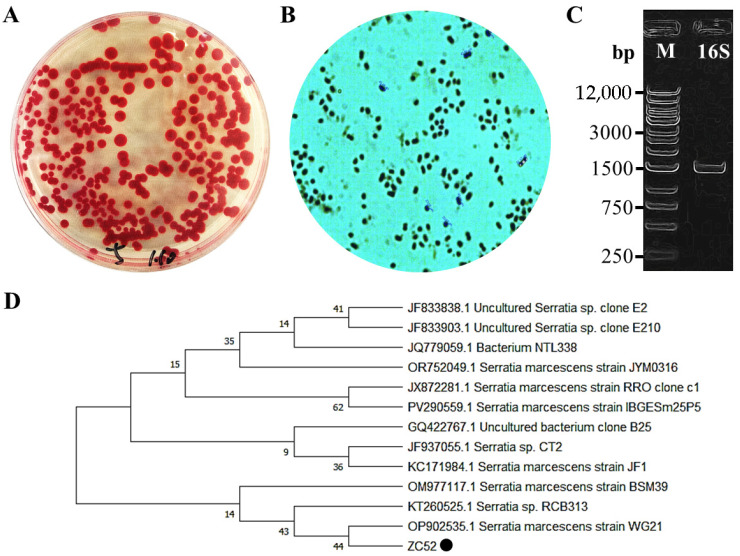
Neighbor-joining tree is used to show the phylogenetic position of *S. marcescens* ZC52 based on the 16s rDNA gene sequence. The numbers at the branches indicate the level of bootstrap support based on the neighbor-joining analysis of 1000 replicates. (**A**) Colony morphology of ZC52 on LB solid medium; (**B**) Microscopic morphology of ZC52; (**C**) Agarose gel electrophoresis analysis of 16S rRNA gene; (**D**) Construction of a phylogenetic tree, with the black dot marking the isolate from this study.

**Figure 2 microorganisms-13-02569-f002:**
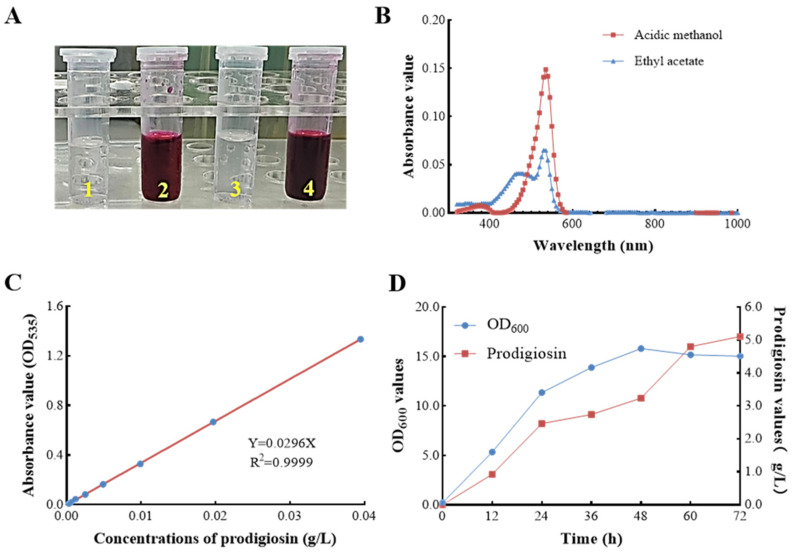
Prodigiosin characterization and preliminary yield evaluation in *S. marcescens* ZC52. The red pigment extracted from the fermentation broth of ZC52 using acidified methanol (left) and ethyl acetate (Right) (**A**), a comprehensive wavelength scan of the pigment extract (**B**), a standard curve plotting the concentrations of crude prodigiosin extracts against absorbance values (**C**), and the kinetics of prodigiosin production by ZC52 (**D**). Values represent the mean of three independent measurements (*n* = 3), and bars indicate SD. All processes were biologically repeated in three independent and parallel experiments.

**Figure 3 microorganisms-13-02569-f003:**
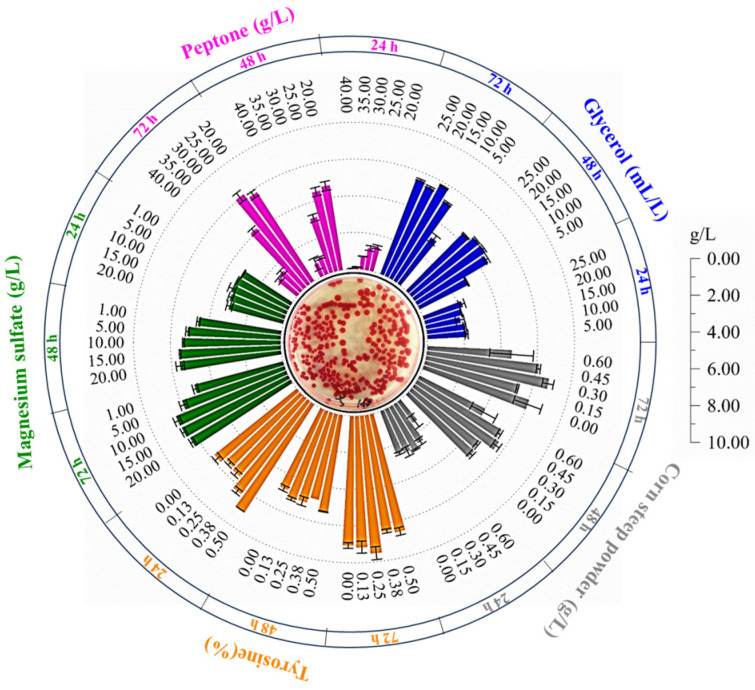
Optimizations of medium composition. Five factors, such as glycerol, bovine bone peptone, magnesium sulfate heptahydrate, corn dry powder, and tyrosine, were systematically investigated to optimize the production performance of prodigiosin by *S. marcescens* ZC52.

**Figure 4 microorganisms-13-02569-f004:**
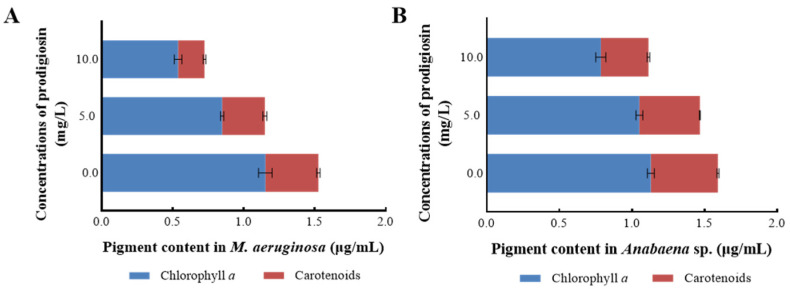
The changes in photosynthetic pigments of *M. aeruginosa* (**A**) and *Anabaena* sp. (**B**) treated with prodigiosin at different concentrations. Values represent the mean of three independent measurements (*n* = 3), and bars indicate SDs. All processes were biologically repeated in three independent and parallel experiments.

**Figure 5 microorganisms-13-02569-f005:**
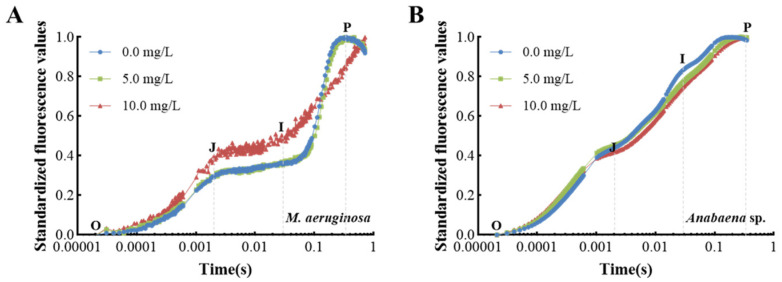
Effect of prodigiosin on the fast induction curves of chlorophyll fluorescence (O J-I-P curve) of *M. aeruginosa* (**A**) and *Anabaena* sp. (**B**). Values represent the mean of three independent measurements (*n* = 3). All processes were biologically repeated in three independent and parallel experiments.

**Figure 6 microorganisms-13-02569-f006:**
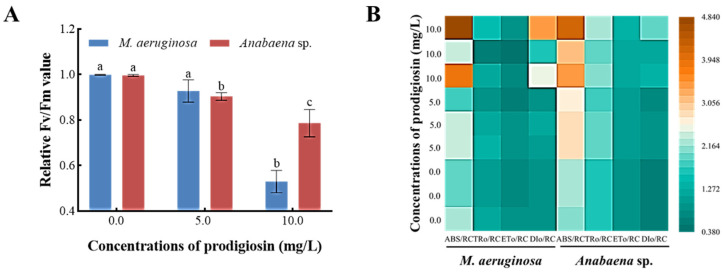
Effects of prodigiosin exposure on the light energy conversion efficiency (**A**) and energy distribution (**B**) in the PSII reaction center of *M. aeruginosa* and *Anabaena* sp. Values represent the mean of three independent measurements (*n* = 3). All processes were biologically repeated in three independent and parallel experiments. Bars with different lowercase letters indicate significant differences among treatments at *p* < 0.05.

**Figure 7 microorganisms-13-02569-f007:**
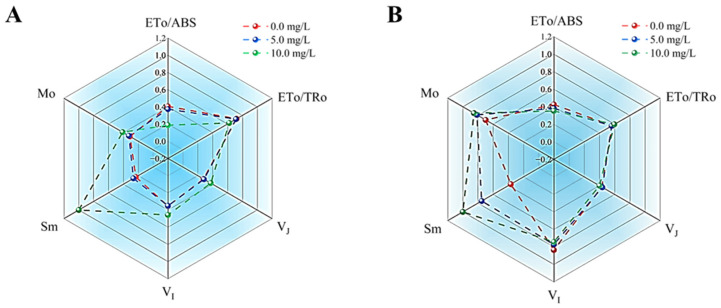
Effects of prodigiosin exposure on the photosynthetic electron transfer efficiency of *M. aeruginosa* (**A**) and *Anabaena* sp. (**B**). Values represent the mean of three independent measurements (*n* = 3). All processes were biologically repeated in three independent and parallel experiments.

**Table 1 microorganisms-13-02569-t001:** Factors and levels of orthogonal design.

Factors	Levels			
	1	2	3	4
A-Glycerol (mL·L^−1^)	5.0	10.0	15.0	20.0
B-Bovine bone peptone (g·L^−1^)	15.0	20.0	25.0	30.0
C-Magnesium sulfate heptahydrate(g·L^−1^)	1.0	5.0	10.0	15.0
D-Corn dry powder (g·L^−1^)	0.00	0.15	0.30	0.45
E-Tyrosine (%)	0.000	0.125	0.250	0.375

**Table 2 microorganisms-13-02569-t002:** L_16_(4^5^) orthogonal design and results.

No.	Factors					Prodigiosin Yield		
	A	B	C	D	E	24 h	48 h	72 h
1	1	1	1	1	1	2.115	2.340	2.404
2	1	2	2	2	2	0.785	1.784	1.871
3	1	3	3	3	3	1.156	2.190	2.254
4	1	4	4	4	4	1.062	1.926	1.997
5	2	1	2	3	4	2.104	3.560	3.706
6	2	2	1	4	3	0.454	1.883	2.064
7	2	3	4	1	2	0.260	1.555	1.373
8	2	4	3	2	1	0.245	0.967	0.920
9	3	1	3	4	2	0.920	2.183	2.707
10	3	2	4	3	1	0.367	0.742	1.058
11	3	3	1	2	4	0.379	2.036	2.313
12	3	4	2	1	3	0.288	2.376	2.644
13	4	1	4	2	3	2.644	6.461	7.645
14	4	2	3	1	4	1.389	3.793	4.132
15	4	3	2	4	1	0.375	1.492	1.772
16	4	4	1	3	2	0.268	1.480	1.654
K1_24 h	1.280	1.946	0.804	1.013	0.776			
K2_24 h	0.766	0.749	0.888	1.013	0.558			
K3_24 h	0.488	0.542	0.927	0.974	1.135			
K4_24 h	1.169	0.466	1.083	0.703	1.234			
Range_24 h	0.792	1.480	0.279	0.310	0.676			
K1_48 h	2.060	3.636	1.935	2.516	1.385			
K2_48 h	1.991	2.050	2.303	2.812	1.751			
K3_48 h	1.834	1.818	2.283	1.993	3.228			
K4_48 h	3.307	1.687	2.671	1.871	2.829			
Range_48 h	1.473	1.949	0.736	0.941	1.843			
K1_72 h	2.130	4.115	2.105	2.635	1.538			
K2_72 h	2.015	2.280	2.498	3.185	1.900			
K3_72 h	2.180	1.925	2.502	2.167	3.647			
K4_72 h	3.797	1.803	3.018	2.135	3.037			
Range_72 h	1.782	2.312	0.913	1.050	2.109			

## Data Availability

The original contributions presented in this study are included in the article. Further inquiries can be directed to the corresponding author.

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
