# Peer review of "Prodigiosin as an Algicidal Agent: Inhibition of Pigment Accumulation and Photosynthetic Efficiency of Cyanobacteria Involved in Algal Blooms"

_microorganisms, 2025, doi:10.3390/microorganisms13112569_

Round 1
Reviewer 1 Report
Comments and Suggestions for Authors
A summary of the paper.
This work presents an important experiment for the production of Prodigiosin, an algaecide with potential for use in controlling blooms of toxic cyanobacteria. The study may be part of a group of methodologies that are ecologically friendly and less impactful on the environment.
My general considerations:
Introduction:
Line 71: The "disruption of algal cell membrane integrity" is mentioned as a characteristic of Prodigiosin. This is considered a dangerous characteristic for an algaecide because cyanotoxin can be released into the water after membrane rupture. How is this characteristic viewed in the analysis of Prodigiosin as a candidate for an algaecide against toxic cyanobacteria?
Line 82: replace Anabaena sp with Dolichospermum sp
Materials and methods:
Replace g/L with g.L-1
Results:
Fig. 1D - Resolution and size can be improved
Fig. 2 could improve the resolution and the legend could contain the information (Fig 2a, 22b, 2c and 2d)
Figs. 6 and 7 need larger fonts and higher resolution of the graphs
Discussion:
Line 422 and 423 – “…including histidine, methionine, and proline—as key building blocks. These compounds, which possess pyrrole-like structures, serve as integral precursors for the synthesis of this pigment.” the Methionine does not have a pyrrole ring. The sentence needs to be rewritten.
Author Response
Reviewer #1:
This work presents an important experiment for the production of Prodigiosin, an algaecide with potential for use in controlling blooms of toxic cyanobacteria. The study may be part of a group of methodologies that are ecologically friendly and less impactful on the environment.
My general considerations:
1)Line 71: The "disruption of algal cell membrane integrity" is mentioned as a characteristic of Prodigiosin. This is considered a dangerous characteristic for an algaecide because cyanotoxin can be released into the water after membrane rupture. How is this characteristic viewed in the analysis of Prodigiosin as a candidate for an algaecide against toxic cyanobacteria?
Response:
Thanks for your suggestion, and we totally agree with it. Prodigiosin exerts a concentration-dependent effect on the disruption of algal cell structure, making it essential to determine its optimal application concentration. This study preliminarily investigated the concentration range capable of effectively inhibiting algal cell proliferation. With subsequent work focusing on the impact of relevant concentrations on the integrity of algal cell membranes, the aim is to achieve effective control of algal growth while minimizing damage to membrane structure.
2)Line 82: replace Anabaena sp with Dolichospermum sp
Response:
Thank you for your valuable comment. Anabaena sp. been reclassified into several genera, with most bloom-forming phytoplankton now attributed to genera such as Sphaerospermopsis and Dilichospermum. In this study, FACHB-82 strain was obtained from the Freshwater Algae Culture Collection of the Institute of Hydrobiology, Chinese Academy of Sciences, where it is catalogued under the genus Anabaena sp. Based on the information available on the official website, its exact taxonomic designation cannot be definitively confirmed. Frankly speaking, we are uncertain whether it should be classified as Dolichospermum sp. Nevertheless, this study will provide a theoretical basis and data support for the toxicological testing of other genera, including Sphaerospermopsis and Dilichospermum.
3)Replace g/L with g.L-1
Response:
Thanks for your suggestion, and we totally agree with it. We have made a correction according to the Reviewer’s comments.
4)Fig. 1D - Resolution and size can be improved
Response:
Thanks for your suggestion, and we totally agree with it. We have made a correction according to the Reviewer’s comments.
5)Fig. 2 could improve the resolution and the legend could contain the information (Fig 2a, 22b, 2c and 2d)
Response:
Thanks for your suggestion, and we totally agree with it. We have made a correction according to the Reviewer’s comments.
6) Figs. 6 and 7 need larger fonts and higher resolution of the graphs
Response:
Thanks for your suggestion, and we totally agree with it. We have made a correction according to the Reviewer’s comments.
7) Line 422 and 423 – “…including histidine, methionine, and proline—as key building blocks. These compounds, which possess pyrrole-like structures, serve as integral precursors for the synthesis of this pigment.” the Methionine does not have a pyrrole ring. The sentence needs to be rewritten.
Response:
Thank you for your valuable comment. We agree with the reviewer that methionine does not possess a pyrrole ring structure, and the sentence has been revised accordingly to ensure accuracy. In the revised manuscript, we clarified that histidine and proline contain pyrrole-like structures and function as structural precursors in prodigiosin biosynthesis, whereas methionine plays a different role by contributing to metabolic regulation and promoting pigment production. The updated text now is highlighted in red. (Lines 443–449). We appreciate the reviewer’s insightful suggestion, which helped us improve the accuracy of our description.
Reviewer 2 Report
Comments and Suggestions for Authors
The paper is focused on important topic and provides interesting decission to mitigate harmful cyanobacterial blooms using algicide produced by a novel strain of Serratia marcescens - a rod-shaped, Gram-negative bacteria from the family Yersiniaceae, a facultative anaerobe and an opportunistic pathogen in humans. The paper is focused on the isolation and identification of bacterial strains capable of producing prodigiosin and optimization of the prodigiosin yield and its large-scale production. The presented results of testing of five factors regarding carbon and nitrogen sources, as well as of inorganic salts as influencing prodigiosin, sound well grounded and promising for future industrial production of this algicide and subseguent maintenance of aquatic systems. Below some more detailed comments are added.
Line 153 - The reviewer completely accepts the name of the strain Anabaena sp. (FACHB-82), as it is provided from the mentioned collection. However, it would be recommended to add one more sentence about this strain (or a short morphological description) because currently Anabaena has been splitted in several genera and most of species that remain in Anabaena are soil algae, while most phytoplankters that cause blooms are from the genera Sphaerospermopsis, Dilichospermum, etc. Such clarification would better orient the reader and future stakeholders.
Line 173 - "concentration of photosynthetic pigments" - better to clarify that this concerns both chlorophyll a and carotenoids since the paper's methods and findings led to protocol for the simultaneous extraction of both chlorophylls and carotenoids.
Line 451 - Microcystis should be in Italic
Line 452 - 'acting as an algicide against Heterosigma akashiwo [13]" - since the reviewed paper is oriented towards harmful cyanobacterial blooms (i.e. caused by non-motile prokaryotic algae from the blue-green evolutionary algal line, mainly from freshwaters), including the information on Heterosigma should be completed by writing that this is a motile, flagellated eukaryotic alga from the yellow-brown algal evolutionary line and is marine representative.
Author Response
Reviewer #2:
The paper is focused on important topic and provides interesting decission to mitigate harmful cyanobacterial blooms using algicide produced by a novel strain of Serratia marcescens - a rod-shaped, Gram-negative bacteria from the family Yersiniaceae, a facultative anaerobe and an opportunistic pathogen in humans. The paper is focused on the isolation and identification of bacterial strains capable of producing prodigiosin and optimization of the prodigiosin yield and its large-scale production. The presented results of testing of five factors regarding carbon and nitrogen sources, as well as of inorganic salts as influencing prodigiosin, sound well grounded and promising for future industrial production of this algicide and subseguent maintenance of aquatic systems. Below some more detailed comments are added.
1)Line 153 - The reviewer completely accepts the name of the strain Anabaena sp. (FACHB-82), as it is provided from the mentioned collection. However, it would be recommended to add one more sentence about this strain (or a short morphological description) because currently Anabaena has been splitted in several genera and most of species that remain in Anabaena are soil algae, while most phytoplankters that cause blooms are from the genera Sphaerospermopsis, Dilichospermum, etc. Such clarification would better orient the reader and future stakeholders.
Response:
Thanks for your suggestion, and we totally agree with it. In this study, FACHB-82 strain was obtained from the Freshwater Algae Culture Collection of the Institute of Hydrobiology, Chinese Academy of Sciences, where it is catalogued under the genus Anabaena. Based on the information provided on the official website, its precise taxonomic designation could not be determined. Nevertheless, this study will provide a theoretical basis and data support for the toxicological testing of other genera, including Sphaerospermopsis and Dilichospermum.
2)Line 173 - "concentration of photosynthetic pigments" - better to clarify that this concerns both chlorophyll a and carotenoids since the paper's methods and findings led to protocol for the simultaneous extraction of both chlorophylls and carotenoids.
Response:
Thank you for your valuable suggestion. We have clarified throughout the manuscript that the analysis of photosynthetic pigments specifically concerns chlorophyll a and carotenoids. For instance, in the methodology section (Line 177-178)
3)Line 451 - Microcystis should be in Italic
Response:
Thank you for pointing this out. We have corrected the formatting, and Microcystis is now presented in italics in the revised manuscript.
4)Line 452 - 'acting as an algicide against Heterosigma akashiwo [13]" - since the reviewed paper is oriented towards harmful cyanobacterial blooms (i.e. caused by non-motile prokaryotic algae from the blue-green evolutionary algal line, mainly from freshwaters), including the information on Heterosigma should be completed by writing that this is a motile, flagellated eukaryotic alga from the yellow-brown algal evolutionary line and is marine representative.
Response:
Thank you for your insightful suggestion. We agree that additional clarification about Heterosigma akashiwo is required for accuracy and better contextualization. Accordingly, we have revised the text to indicate that H. akashiwo is a motile, flagellated eukaryotic algae and is a representative of harmful marine bloom-forming species. In addition, we have further emphasized that this example highlights the broad-spectrum algicidal activity of prodigiosin, which is capable of inhibiting both freshwater bloom-forming cyanobacteria (e.g., Microcystis) and marine eukaryotic microalgae such as H. akashiwo. This information strengthens the rationale for investigating prodigiosin-based strategies for bloom mitigation. The corresponding revision has been incorporated into the manuscript. (Line 475-480).